# Experiment and Simulation Study of the Laser-Induced Cavitation Bubble Technique for Forming a Microgroove in Aluminum Foil

**DOI:** 10.3390/mi14112106

**Published:** 2023-11-17

**Authors:** Liangliang Wang, Chun Su, Xiaofeng Jia, Zhongning Guo, Zhixiang Zou

**Affiliations:** 1School of Aeronautics and Mechanical Engineering, Changzhou Institute of Technology, Changzhou 213032, China; wangll99@163.com (L.W.); suc@czu.cn (C.S.); 2School of Mechanical Engineering, Anyang Institute of Technology, Anyang 455000, China; jiaxf@ayit.edu.cn; 3State Key Laboratory of Precision Electronic Manufacturing Technology and Equipment, Guangdong University of Technology, Guangzhou 510006, China; znguogdut@163.com

**Keywords:** laser, cavitation bubble, micro-forming, axisymmetric structure, thickness thinning, hardness

## Abstract

The present work introduces a laser-induced cavitation bubble technique for forming an axisymmetric structure (i.e., microgroove) and the dynamics of a cavitation bubble from initial expansion to the collapse stages that were also simulated. Furthermore, the shock wave signals and dynamic properties of the cavitation bubble were recorded using a hydrophone and a high-speed camera. The experiments on microgrooves formed by laser-induced cavitation bubble stamping were carried out, and the effects of laser energy, the initial position of the bubble, and the number of impacts on the microformability of aluminum sheets are discussed. The depth of the microgroove was investigated using experiments, and it was found that the process can serve as a rapid technique for impressing microfeatures on thin-sheet metals. The experimental results showed that as the initial position of the bubble increased, the deformation depth decreased. As the laser energy and number of impacts increased, the deformation depth increased. The results of the response surface experiments showed that a laser energy of 27 mJ, 3 impacts, and a bubble position of 3 mm were optimal for the process. By using the optimal parameters, flat and smooth microgrooves with a forming depth of 102.54 µm were successfully fabricated. Furthermore, the maximum thickness thinning of the microgroove section occurred at the entrance areas, and this area had the greatest hardness. This also indicated that the greatest amount of plastic deformation of the material and grain refinement occurred in this area. On the other hand, the aluminum foil did not undergo oxidation during the plastic deformation process. These results demonstrated that laser-induced bubble stamping is an advanced micromachining method with promising applications.

## 1. Introduction

As the demand for microdevices increases with the rapid development of electronics, medical devices, and micro-electro-mechanical systems, micro-forming technology in association with micrometal parts is becoming increasingly significant [1,2,3]. However, conventional metal micro-forming processes are subject to punch wear, which has been a challenge in forming accuracy [4]. Additionally, the “scaling effect” caused by miniaturization reduces the formability of metals and introduces tremendous difficulties in mold manufacturing, which has become a crucial bottleneck for conventional micro-forming techniques [5]. Accordingly, to solve the above-mentioned critical issues in metal micro-stamping, considerable investigations have been conducted [6,7].

Vollertsen et al. [8] observed that by using laser-induced shock waves instead of micropunches during the forming process, the wear of micropunches can be fundamentally eliminated. Nevertheless, in laser-induced shock processes, additional serious ablation problems are commonly involved. For this reason, to prevent the problem of ablation on the surface of the workpiece, the technique of applying a sacrificial layer such as aluminum foil to the workpiece surface was proposed [9]. For instance, Zhang et al. [10] adopted laser irradiation to produce shock waves on aluminum alloy plates coated with black tape, and the laser-generated shock effect was used to form conical cup structures on the metal plates and decrease the ablation behavior. However, under multiple impacts, the sacrificial layer on the workpiece surface is not available for re-application. Apparently, it is necessary to repeatedly adhere the sacrificial layer, thereby seriously reducing the efficiency of the actual forming process. As a consequence, it was also suggested that the sacrificial layer could be replaced by an elastic material for reusability in laser-induced stamping processes [11]. For example, Wang et al. [12] reported a new technique for fabricating thin metal foil micro-parts by laser shock wave with a forming/punching composite mold, where a flexible rubber material was used as a soft punch acting on the thin metal sheet. Moreover, it was also experimentally found that the ablation state of the enclosed medium and the surface roughness of different regions varied with energy. Also, suitable energy is necessary to form complex parts, and the molding process can be applied to make parts with good surface quality [13]. Nonetheless, when using elastic materials, the impact pressure acting on the workpiece is greatly weakened, which severely affects the forming efficiency [14]. This problem has not been fundamentally solved.

On the other hand, liquid mediums (i.e., water) are also used instead of conventional elastic materials. More recently, Zhong et al. [15] observed that lasers not only break down liquids but also produce plasma shock waves and cavitation bubbles. Consequently, underwater laser-induced bubble technology for stamping was proposed. Ren et al. [16,17] investigated the process of cavitation shot peening to strengthen the alloy. Takada [18] used a cavitation bubble to create a circle with a depth of 30 μm on the surface of titanium metal. Nie et al. [19] utilized a nanosecond laser to break down water, and a metal was deformed underwater by a cavitation bubble. Dular et al. [20] observed the collapse of a bubble in the liquid and discovered that the microjet mechanism caused the majority of the damage. Chen et al. [21] proposed stamping aluminum foil using the laser-induced cavitation bubble technique, and a circle with a depth of 167.2 μm was formed on 10 μm thick aluminum foil. Although there has been research on the production of centrosymmetric structures such as micro-pits and micro-holes, the axisymmetric structure (i.e., groove) has received little attention [22]. On the other hand, thickness thinning is an essential evaluation criterion for plastic deformation of metal sheets [23,24]. In general, when plastic deformation of the material occurs, severe thinning leads to stress concentration, local necking, and even part failure [25,26]. More recently, in the laser-induced cavitation bubble formation process, thickness thinning has been extensively investigated [27,28]. Nonetheless, the thickness thinning of the axisymmetric structure (i.e., groove) has been poorly investigated [29]. Furthermore, it should be noted that the thickness thinning of structures plays a role in their hardness. However, in the laser-induced cavitation bubble formation of the axisymmetric structure process, only a few published papers that investigate the relationship between the thickness thinning and the hardness can be found in the literature. As a consequence, it is an urgent issue that requires more attention and investigation.

Inspired by this, the present work used both experimental and numerical methods to investigate the evolution of a bubble oscillating near the guide hole. The experiments on microgroove formation by laser-induced cavitation bubble stamping were conducted under varying laser energy, number of impacts, and initial positions of the bubble. Furthermore, the present study also further investigates the relationship between thickness thinning and hardness to understand its mechanisms.

## 2. Experiment Designs

### 2.1. Experimental Setups

The laser-induced cavitation bubble stamping process is depicted schematically in Figure 1. To create cavitation bubbles, a Gaussian Q-switched Nd:YAG laser with a wavelength of 532 nm and a pulse width of 7 ns was employed. A guide hole with a diameter of 600 μm was used to constrict the bubble and guide the pressure to the workpiece. The growth and change processes of bubbles were captured by a high-speed camera (FASTCAMSA-Z, Tokyo, Japan) with a maximum frame rate of 100,000 frames per second. An LED light source with a parallel beam illuminated the cavitation bubble region. During the development of the cavitation beneath the water, a hydrophone recorded variations in impact pressure. The oscilloscope sampling frequency was set at 100 MHz, and the hydrophone was calibrated at 1–35 MHz. The initial position of the cavitation bubble is adjusted with 0.02 mm precision by modifying the three-dimensional movable platform. The pressure generated by the shock waves and the microjet acts on the workpiece through the guide hole. The workpiece developed a plastic deformation to replicate the cavity shape of the die under the pressure of the shock waves and the constraint of the fixed die. Figure 2 shows a microgroove mold made by electric discharge machining on the surface of stainless steel with a width of 240 μm and a depth of 105 μm.

### 2.2. Materials and Methods

Aluminum (Al) is widely used in micro-electro-mechanical devices, vehicles, and other industries because of its soft texture, good ductility, and abrasion resistance. Aluminum foils 20 μm thick were chosen for the experiments. Before the experiments, the aluminum foil was first cut into squares of 8 × 8 mm. Then, the surface was cleaned with ultrasonic alcohol for 3 min. A laser confocal microscope (OLS4100, Olympus, Tokyo, Japan) was employed to measure the depth and width of the microgroove created by laser-induced cavitation bubble impact. The surface morphology is studied by using a scanning electron microscope (SEM) (S-3400N, Hitachi, Japan, Tokyo), and the element is analyzed by energy dispersive spectroscopy (EDS). Furthermore, the nanoindentation equipment (NHT-TTX2, Anton Pa, Zurich, Switzerland) determined the hardness.

## 3. Numerical Simulations

### 3.1. Simulation Model

The initial cavitation is assumed to be spherical, and the water is a Newtonian liquid that is incompressible, in order to calculate the impact pressure created by the bubble. A 2D half model was used to improve the computation efficiency (see Figure 3a). The pressure-out border was set to DE, the symmetry center boundary was set to OE, and the walls were set to OA, AB, BC, and CD. The center of the bubble is on the symmetric axis, and the distance from the guide hole entrance is H. Assume that the flow is laminar and that the pressure at the DE boundary is equal to 1 bar. As shown in Figure 3b, the entire area is divided by a quadrilateral grid with a total of 126,909 grids.

### 3.2. Governing Equations

The pressure-based type solver was utilized in Fluent, and the duration for transient analysis was specified, as well as the volume of fluid (VOF) model for two phases, water and gaseous vacuoles. It is assumed that in the changing cycle of the cavitation bubble, the bubble contains solely water vapor, ignoring gravity and mass. The vapor was considered to be the ideal gas, and the pressure equation was PRESTO-style. The VOF model was utilized to track the interaction between the bubble and the liquid during the simulation. The governing equation of the fluid in simulation is expressed as follows:(1)$$\rho =\rho_{l}\alpha_{l}+\rho_{g}\left(1-\alpha_{l}\right)$$
(2)$$\mu =\mu_{l}\alpha_{l}+\mu_{g}\left(1-\alpha_{l}\right)$$
where $\alpha_{l}$ is the volume fraction of the liquid phase, $\rho_{l}$ is the density of liquid, $\rho_{g}$ is the density of gas, $\mu_{l}$ is the viscosity of liquid, and $\mu_{g}$ is the viscosity of gas. 

The equation for the liquid volume fraction is:(3)$$\frac{\partial }{\partial t}\alpha_{l}+\nabla \left(\alpha_{l}\cdot \vec{v}\right)=0$$
where $\vec{v}$ is the velocity of the liquid phase. Through a first-order explicit technique, the governing equations were solved using the finite volume method with a linked coupled pressure-based algorithm. The energy equation was solved using the second-order upwind approach. Using the PRESTO! scheme to discretize the pressure equation and the PISO algorithm for pressure-velocity coupling, the time step was set to 1 × 10^−8^ s and 50,000 s, respectively. 

### 3.3. Initial Conditions

The temperature and pressure of the initial bubble must be established before the dynamic features of the cavitation bubble can be simulated. The initial internal energy (*E*) of a cavitation bubble is:(4)$$E=E_{0}+\Delta E$$
where $E_{0}$ is the initial internal energy of the distilled water, Δ*E* is the energy absorbed by water, and the absorption of water is 2.1% [21]. The mass of the initial bubble is calculated as follows:(5)$$M=\frac{4}{3}\pi R_{0}^{3}\rho_{0}$$
where $\rho_{0}$ is the initial density of gas and $R_{0}$ is the initial bubble radius. 

The ideal gas equation is:(6)$$E=\frac{i}{2}PV=\frac{i}{2}MRT$$
where *V* is the specific volume of the gas, *i* is the gas degree of freedom (*i* = 6), and *R* is the commonly used specific ideal constant of gas for water steam (*R* = 461.5 J/Kg·K^−1^). Combining with Equations (4)–(6), the initial pressure and temperature of the cavitation bubble can be calculated as follows:(7)$$P=\frac{2E}{iV}$$
(8)$$T=\frac{2E}{iMR}=\frac{3E}{2\pi iRR_{0}^{3}\rho_{0}}$$

As a result, the initial pressure and temperature within a bubble are 214 MPa and 4417 K, respectively.

## 4. Results and Discussion

### 4.1. Bubble Dynamic Characteristics

In the experiment, the value of *H* is 2 mm, and the dynamic features of a cavitation bubble with a laser energy of 19 mJ are shown in Figure 4. At 0 μs, the laser breaks down the water, generating high-temperature and high-pressure plasma. Following that, the plasma absorbs the laser and swells outward, forming an initial bubble. At 110 μs, the bubble swells to its maximum diameter of 2.1 mm. The cavitation bubble then shrinks due to the presence of the wall, and the bubble’s upper and bottom walls degrade at different rates. It eventually collapses at 240 μs, producing a microjet directed at the guiding hole. 

The results of the simulation for the cavitation bubble oscillation period are shown in Figure 5a. It takes 116 μs from bubble creation to a maximum diameter of 2.18 mm. The modeling result for the maximum cavitation bubble radius is greater than the experiment result. The real time it takes for a bubble to collapse is 240 μs, although the numerical duration is 245 μs. The results of the cavitation bubble simulation are essentially consistent with the experimental data, and the error is within an acceptable range, confirming the model’s accuracy. The radius of growth of the laser-induced cavitation bubble in the experiment and simulation is depicted in Figure 5b. The numerical outcome of the bubble’s growing and collapsing process is found to be quite close to the experimental result. It can be considered that the computational result’s water jet velocity is qualitatively and quantitatively identical to the experimental finding [6,27]. Figure 5c depicts the acoustic signals detected by a hydrophone. The hydrophone is positioned vertically, 2 mm below the cavitation bubble on the workpiece. There are two primary peaks, which reflect the plasma shock wave and the cavitation bubble collapse shock wave, respectively. The two peak signals have values of 0.52 V and 0.23 V, corresponding to pressures of 26 MPa and 11.5 MPa, respectively.

Figure 6 depicts the simulation results of microjet velocity when the bubble is close to the guide hole. The cavitation flow velocity is nearly zero during the process of cavitation bubble expansion. When the cavitation bubble collapses, the velocity field of the microjet is visible on top of the rigid wall, like a tiered mushroom cloud. From the top, the microjet enters the cavitation bubble and is focused on the center, which shoots towards the guidance hole. The highest jet velocity can approach 104 m/s, and the corresponding impact pressure can be estimated using Formula (9).
(9)$$P=\frac{\rho_{1}C_{1}\rho_{2}C_{2}}{\rho_{1}C_{1}+\rho_{2}C_{2}}V$$

Because the parameter value for the solid phase is significantly greater than the value for the liquid phase ($\rho_{1}C_{1}\ll \rho_{2}C_{2}$), it can be simplified as:(10)$$P=\rho_{1}C_{1}V$$
where *P* is the impact pressure, $\rho_{1}$ and $C_{1}$ are the density and sound velocity of water ($\rho_{1}$ = 998 kg/m^3^, $C_{1}$ = 1483 m/s), respectively, $\rho_{2}$ and $C_{2}$ are the density and sound velocity of 304 stainless steel, respectively, and *V* is the jet speed.

Figure 7 shows the temperature field distribution simulation results as well as the change in temperature value. It can be seen that the temperature rapidly drops from 1300 K during the initial expansion of the bubble, then sharply rises to 964 K, and then rapidly falls when the bubble collapses. Because of the Bjerknes force, the cavitation bubble collapses in the direction of the guiding hole. It should also be noted that the bubble’s greatest temperature appears in the center of the bubble rather than on the bubble wall.

### 4.2. The Effect of the Initial Position of the Bubble (H) 

Figure 8 depicts the profile of a microgroove generated by the impact of different bubble positions on a 20 μm aluminum foil. The values of H are +1 mm, +2 mm, +3 mm, and +4 mm, respectively. The energy of the laser is 23 mJ and only impacts the aluminum foil once. As seen in Figure 8a,b, there are various-sized spots at the bottom of the microgroove, and the area of the spot decreased from 4.35 mm^2^ to 1.6 mm^2^ as the values of *H* increased. When the values of *H* increased from +1 mm to +4 mm, the forming depth of the microgroove decreased from 100.19 μm to 50.23 μm, and the width was approximately 240 μm, as shown in Figure 8c,d. As the values of H increase, the impact pressure and irradiation temperature on the surface of the workpiece weaken, so the forming depth of the workpiece decreases and the ablation area decreases. 

Figure 9 shows the surface morphology of a microgroove with a SEM. There is flocculent spatter on the surface of the workpiece in the square blue area on the surface of the microgroove, as shown in Figure 9a. This is attributed to the high temperature causing the workpiece to melt and vaporize. The vaporized melt undergoes flow under the influence of high-pressure plasma, forming a remelted layer on the surface after condensation and solidification [30]. The high-temperature melted aluminum foil may also chemically react with oxygen in the liquid, resulting in a thin oxide layer on the surface of the remelted layer [31]. As shown in Figure 9b,c, the oxygen content has a significant difference between the original and processed surfaces (0.761% and 9.027%, respectively). It indicates that certain areas of the aluminum foil’s surface have been oxidized during the process. The oxidation of aluminum may be caused by the dissociation of water molecules during laser focusing, which releases oxygen that interacts with the aluminum foil surface to form aluminum oxide.

### 4.3. The Effect of Laser Energy

Figure 10 depicts the forming profile of 20 μm Al foil thickness under varying laser energy (19 mJ, 23 mJ, 27 mJ, and 31 mJ). With an *H* value of 3 mm, the laser impacts the workpiece in a single shot. As shown in Figure 10a,b, the ablation area of the microgroove surface increased from 1.66 mm^2^ to 3.6 mm^2^ as laser energy increased. Because increased laser energy causes the temperature inside the cavitation bubble to grow, the ablation region will begin to expand. Figure 10d shows the cross-sectional curve profile at the maximum forming depth under laser energy. The depth of the microgroove increased from 42.3 μm to 103.6 μm, and the width increased from 228 μm to 238 μm, as shown in Figure 10e. The forming depth of Al foil becomes stable as the laser energy increases. Because high laser energy produces a stronger plasma shock wave and a larger cavitation bubble diameter, the impact pressure increases, thus increasing the forming depth of the workpiece [32]. When the forming depth of the workpiece is close to the depth of the mold, the forming depth tends to be stable even when the laser energy is increased. Furthermore, the punched microgroove is not uniformly formed; the middle is deep, and the two sides are shallow. This is because the vertical wall of the die confines the workpiece as it is driven into the slot die under impact pressure, causing the workpiece’s center to be deeper than the end, as shown in Figure 10c.

### 4.4. The Effect of the Number of Impacts

Figure 11 depicts laser impact at various times, which can result in different deformation results. The number of impacts is 1, 3, and 5, respectively. The value of *H* is 4 mm, and the laser energy is 19 mJ. As demonstrated in Figure 11a,b, a microgroove structure is formed on the aluminum foil’s surface with no surface ablation. When the number of impacts increased from 1 to 5, the depth of the microgroove increased from 41.6 μm to 103.5 μm, while the width changed only slightly and remained relatively constant at 240 μm, as shown in Figure 11c,d. According to the analysis, the number of impacts has a significant influence on the formation depth. However, as the number of impacts increases, the formation depth becomes saturated. This is because when the number of impacts increases, the value of *H* increases due to the concave plastic deformation of aluminum foil, causing the impact pressure to decrease. Furthermore, as the number of impacts increases, the forming depth approaches the mold depth, and the final forming depth of the microgroove reaches the mold depth, which tends to be a stable value. 

### 4.5. Response Surface Methodology to Optimize the Forming Depth

The parameters of the forming depth are optimized by using the composite design principle of the response surface methodology. In the experiment, the laser energy, the initial position of the bubble, and the number of impacts are identified based on Box Behnken’s central combination experimental design concept, as illustrated in Table 1. The Design Expert develops 17 sets of test points in accordance with the variables and response values, as shown in Table 1.

Regression analysis was performed using Design Expert 12 software on the data in Table 2, and the results are given in Table 3. When *p* < 0.0001, it indicates that the regression equation model is highly significant. The data are subjected to multiple quadratic response surface fittings by the Design Expert, and the following quadratic regression equation model of forming depth is obtained:(11)$$\begin{array}{l} Y=107.94-23.56X_{1}-1.84X_{2}+24.93X_{3}+0.44X_{1}X_{2}+4.88X_{1}X_{3}-0.83X_{2}X_{3}-1.56X_{1}^{2}+\\ 0.08X_{2}^{2}-2.05X_{3}^{2} \end{array}$$

Figure 12a depicts the normal probability distribution of the formation depth by laser-induced cavitation impact; the points in the picture are closer to a slash, indicating a better fit. Furthermore, in Figure 12b, the reliability analysis of the quadratic regression equation of the formation depth shows that the actual experimental values are closer to a slash, indicating that the projected model and experiment fit well and that the model is feasible.

The effects of the factor interaction on the forming depth are shown in Figure 13. According to the optimized analysis by the Design Expert, the best optimization parameter conditions are as follows: laser energy is 27 mJ; number of impacts is 3; bubble position is 3 mm; best optimization forming depth is 102.54 µm.

### 4.6. The Surface Morphology

Using response surface optimization settings, laser-induced cavitation bubble impact formation experiments were carried out, and Figure 14 depicts the surface profile formed in a microgroove. The surface of the aluminum foil produces a microgroove, and there are no flaws such as ablation or oxidation on the bottom surface of the microgroove, as shown in Figure 14a. The optimized parameters were utilized to impact the surface of the aluminum foil, and the processed microgroove depth reached 103 µm, which was close to the mold depth, as illustrated in Figure 14b.

When scanning electron microscopy (SEM) is used to observe the surface of the microgroove, it can be seen that the microgroove is flat and smooth, with no oxidation, as shown in Figure 15a. The surface element content of the microgroove was determined by analyzing the bottom portion of the microgroove using X-ray energy dispersion spectroscopy, as shown in Figure 15b. Due to the use of aluminum foil in the experiment, in order to determine whether oxidation reactions occur during the deformation process of the microgroove, the content of aluminum and oxygen elements on the surface of the microgroove was extracted. It can be seen that the oxygen content is only 0.167%, indicating that the aluminum foil did not undergo oxidation during the plastic deformation process.

## 5. Thickness Distribution and Hardness Analysis

In general, when aluminum foil is subjected to plastic deformation under impact force, the thickness thinning is a significant criterion for evaluating sheet metal, and as a consequence, the local fractures will occur under the condition of severe thinning [33]. It is important to note that the thickness thinning of structures plays a role in their hardness [29]. Thus, this section studied the relationship between thickness thinning and hardness to understand its mechanisms. In the present investigation, the produced microgroove is fixed by cold embedding with low-viscosity epoxy resin and grinding with 240 # to 2000 # sandpaper. The laser confocal microscope was used to determine the cross-sectional thickness of the generated microgroove, and the measurement result is shown in Figure 16. It can be seen that the thickness of the generated microgroove section has decreased to variable degrees at different points when compared to the initial aluminum foil thickness. Under the action of the laser impact force, the aluminum foil bends, stretches, and flows into the mold cavity, lowering its thickness. As the impact force received at various points varies, the thickness decrease in the cross-section of the microgroove differs at different positions. The maximum thinning of the microgroove section occurs at the entrance of the mold (positions 2 and 8). This is because when aluminum foil is subjected to the instantaneous impact of laser-induced cavitation, the contact friction between the mold and the aluminum foil increases the flow resistance of the sheet metal, making it difficult for the aluminum foil to flow into the mold cavity [29]. As a consequence, the material matching the mold cavity is stretched and bent by the impact force, resulting in the aluminum foil near the entrance to the mold being thinned the most severely.

The surface hardness of the formed microgroove is measured using a nano-indentation hardness tester. During the measurement, when a force is applied to the surface of the workpiece through the nano-indentation hardness tester, the workpiece undergoes elastic deformation due to the load applied to its surface. The hardness of various locations of the produced microgroove section was tested, and the variations in surface loading and indentation depth of the aluminum foil are shown in Figure 17. It can be observed that the maximum load applied to the workpiece is 10 mN for a duration of 10 s. The loading and unloading rates are both adjusted to 10 mN/min, as illustrated in Figure 17a. Furthermore, it should be noted that the surface of the aluminum foil does not change in the depth of indentation into the material by the indenter of the nano-hardness tester during the holding time. In contrast, during load unloading, the workpiece has some rebound in the depth of indentation of the indenter due to the presence of elastic deformation, as shown in Figure 17b.

Figure 18 shows the variation in the nano-hardness values measured at the formed microgroove cross-section, and different cross-sectional locations were selected for the experiments. Figure 18a shows the local distribution of the nano-indentation loading locations, which indicates that there are obvious indentation marks on the surface of the aluminum foil. Additionally, Figure 18b shows the measured hardness value change curve of the aluminum foil after plastic deformation by impact, and it can be seen that the initial hardness value of the aluminum foil is 1019 MPa. Though the hardness value increases when the material is plastically deformed by the impact force, the hardness value changes differently for different locations of the material. For instance, the hardness value of the microgroove cross-section where the thickness thinning is greatest (positions 2 and 8) was increased to 1900 MPa, an increase in hardness value of 86.4%. This also indicated that the greatest amount of plastic deformation of the material occurred in this area, meaning that the hardness value increased the most [34].

## 6. Conclusions

In this paper, a microgroove-forming method based on laser-induced cavitation bubbles is proposed. Laser-induced cavitation and impact aluminum foil formation have been investigated through experiments and numerical models. Furthermore, the feasibility of microgrooves formed by laser-induced cavitation bubble stamping has been demonstrated. Some conclusions can be drawn from this paper:The energy dispersive spectroscopy analysis of the oxygen content on the bottom of the microgroove revealed that oxidation reactions occurred after the Al foil was processed, and the oxygen content increased from 0.761% to 9.027%.The depth of the aluminum foil surface microgroove development increased from 42.3 μm to 103.6 μm, and the laser energy (number of impact) increased from 19 mJ to 31 mJ (1 to 5 times). The microgrooves on the surface of aluminum foil have different depths, with the forming depth in the middle being greater than at both ends.The results of the response surface experiments showed that to fabricate microgrooves using the laser-induced cavitation bubble technique, the optimal process conditions were as follows: laser energy is 27 mJ, the number of impacts is 3, and the bubble position is 3 mm.By employing an optimal parameter, the flat and smooth microgroove with a forming depth of 102.54 µm was successfully fabricated.The maximum thickness thinning of the microgroove section occurred at the entrance area, and this area had the greatest hardness. This also indicated that the greatest amount of plastic deformation of the material occurred in this area.

## Figures and Tables

**Figure 1 micromachines-14-02106-f001:**
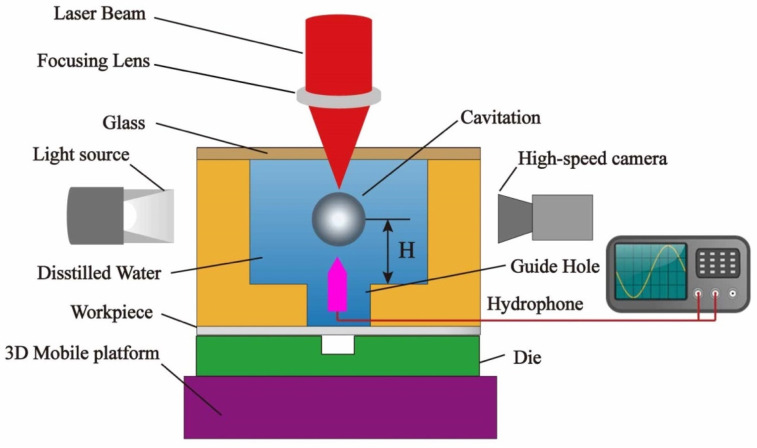
Schematic diagram of the experimental device.

**Figure 2 micromachines-14-02106-f002:**
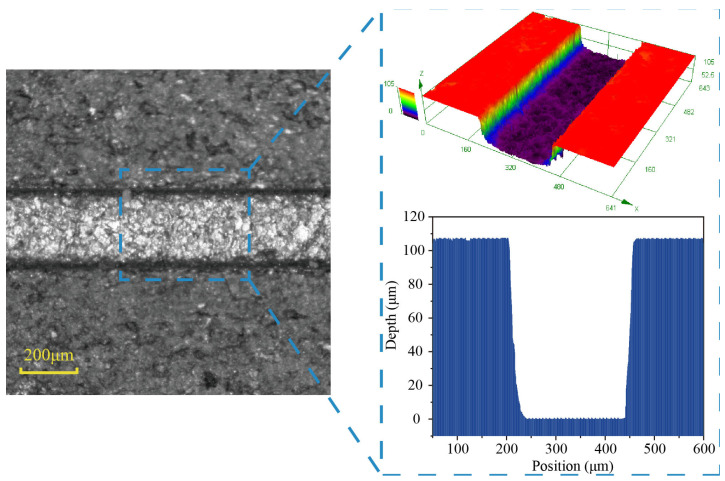
Mold of the microgroove.

**Figure 3 micromachines-14-02106-f003:**
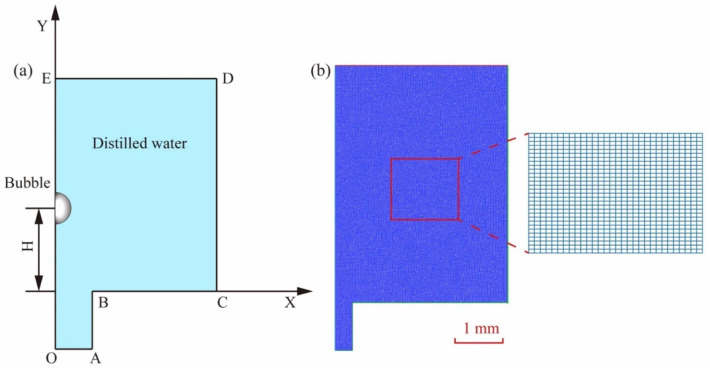
(**a**) Cavitation bubble boundary conditions. (**b**) Mesh gridding.

**Figure 4 micromachines-14-02106-f004:**
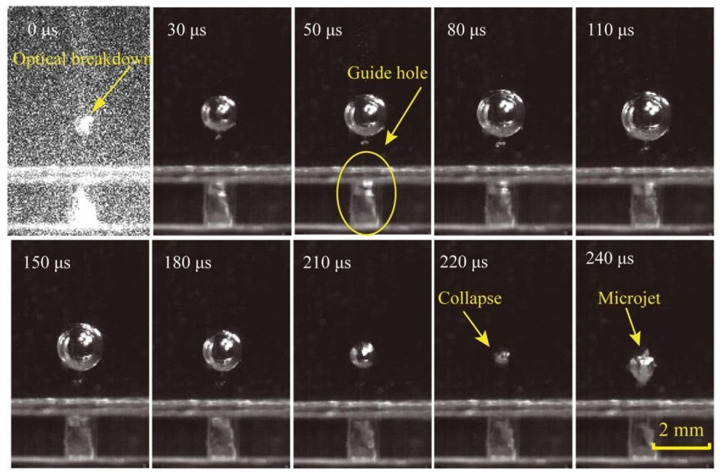
High-speed camera images of the evolution of a cavitation bubble.

**Figure 5 micromachines-14-02106-f005:**
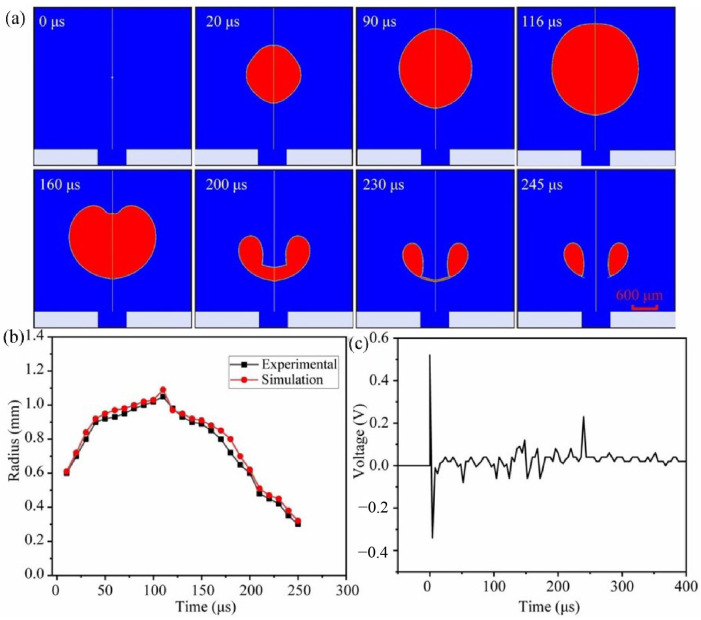
(**a**) Simulation of the bubble pulsation sequence. (**b**) Evolution of the laser-induced cavitation bubbles. (**c**) Curve of the acoustic signals of the cavitation bubble.

**Figure 6 micromachines-14-02106-f006:**
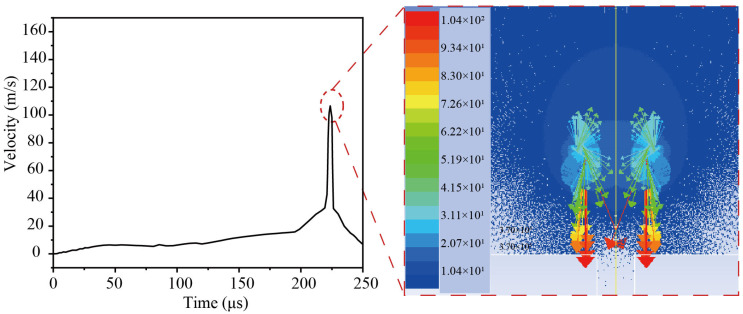
Velocity curve of the microjet simulation.

**Figure 7 micromachines-14-02106-f007:**
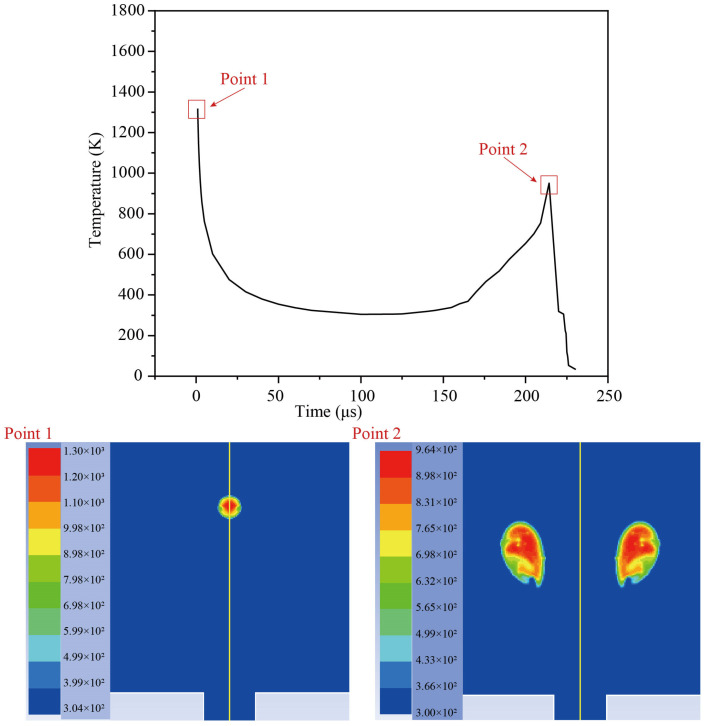
Change in temperature (K) in the pulsation of the cavitation bubble.

**Figure 8 micromachines-14-02106-f008:**
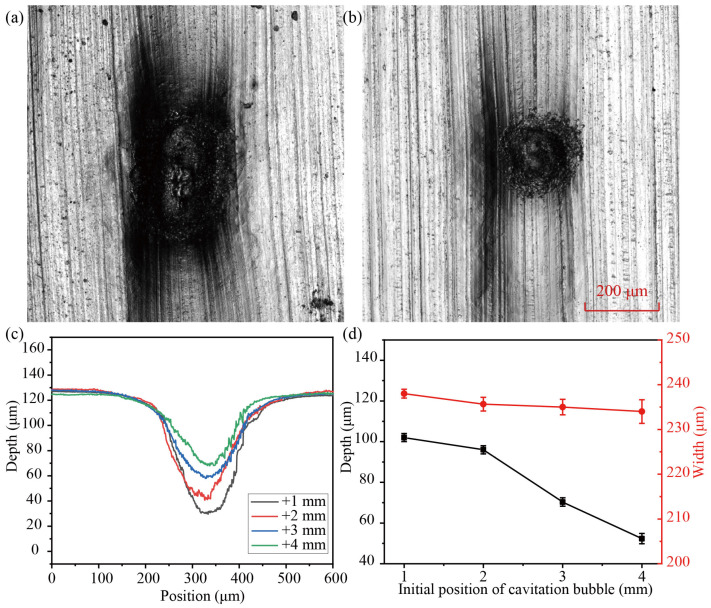
Surface morphology of the aluminum foils at (**a**) +1 mm and (**b**) +4 mm. (**c**) Section profiles of microgrooves for different positions of the bubble. (**d**) Influence of the initial positions of the bubble on deformation depth and width.

**Figure 9 micromachines-14-02106-f009:**
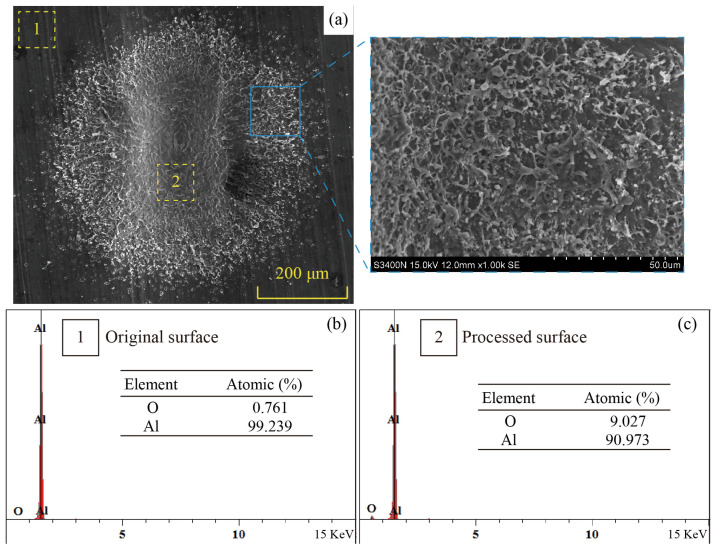
(**a**) SEM image of the foil surface. (**b**,**c**) Oxygen content of Al foil before and after processing.

**Figure 10 micromachines-14-02106-f010:**
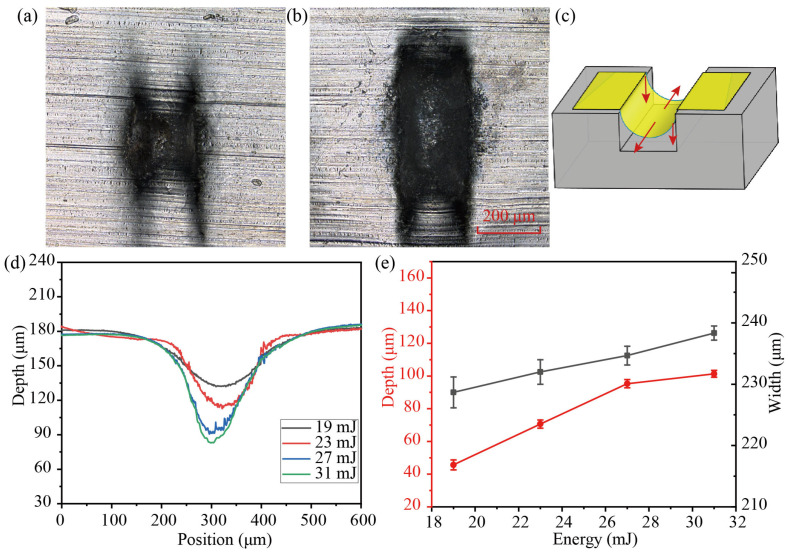
Surface morphology of aluminum foils with varying laser energy: (**a**) 19 mJ and (**b**) 31 mJ. (**c**) Curve profiles of the microgroove under varying laser energy. (**d**) Influence curve of laser energy on forming depth. (**e**) Influence of laser energy on deformation depth and width.

**Figure 11 micromachines-14-02106-f011:**
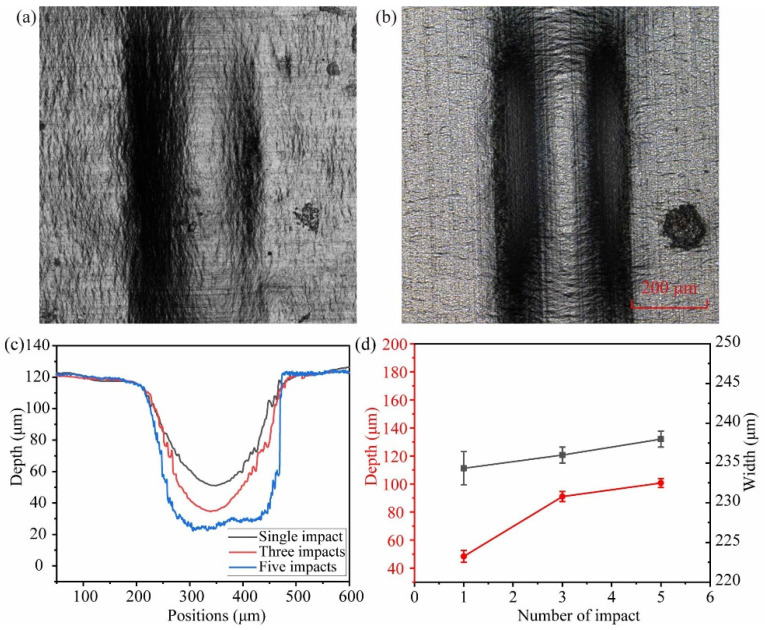
Surface morphology of aluminum foils with different impact times: (**a**) 1 and (**b**) 5. (**c**) Depth of the microgroove section under different impact times. (**d**) Influence of the number of impacts on the depth and width of deformation.

**Figure 12 micromachines-14-02106-f012:**
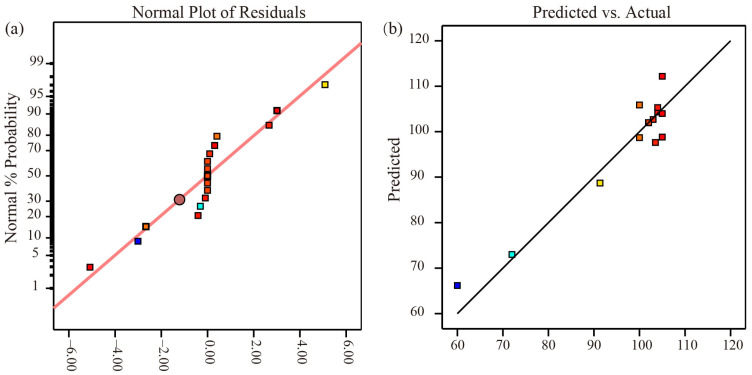
Model reliability analysis diagram. (**a**) The normal probability distribution of the formation depth; (**b**) The reliability analysis of the quadratic regression equation.

**Figure 13 micromachines-14-02106-f013:**
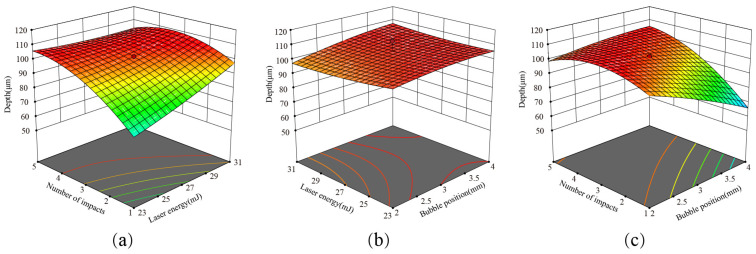
Effect of factor interaction on the forming depth. (**a**) The effect of number of impacts and laser energy on depth; (**b**) The effect of laser energy and bubble position on depth; (**c**) The effect of number of impacts and bubble position on depth.

**Figure 14 micromachines-14-02106-f014:**
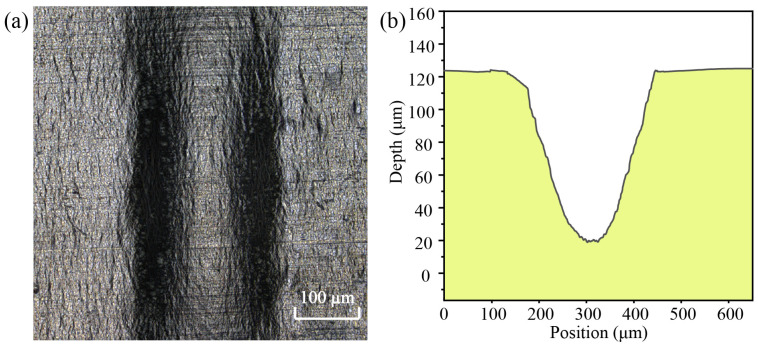
Surface of the forming microgroove. (**a**) Surface of the microgroove; (**b**) Microgroove section depth curve.

**Figure 15 micromachines-14-02106-f015:**
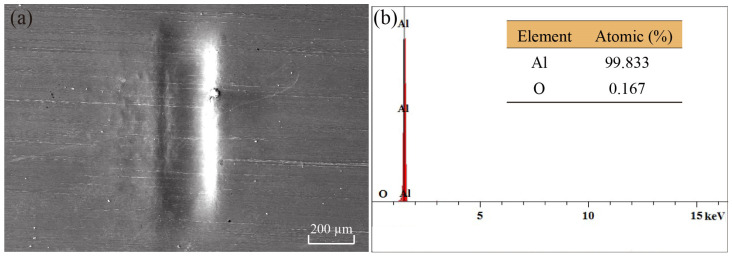
EDX surface scanning analysis of the microgroove. (**a**) Surface of the microgroove; (**b**) Element content of oxygen and aluminum on the surface of microgroove.

**Figure 16 micromachines-14-02106-f016:**
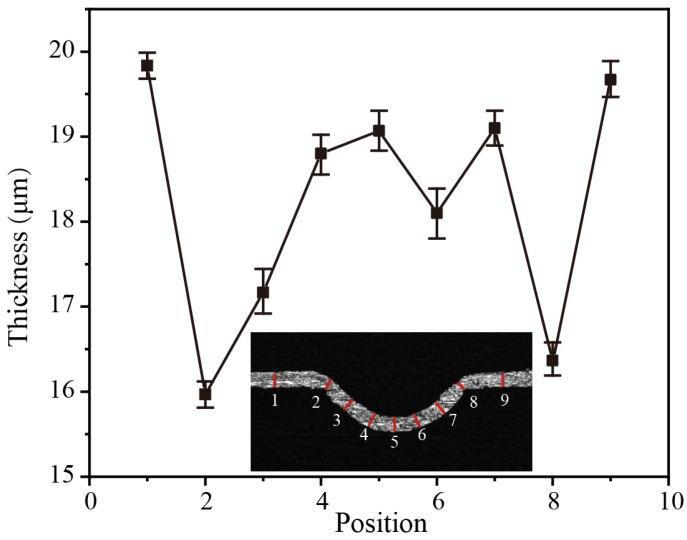
Thickness distribution of the microgroove section.

**Figure 17 micromachines-14-02106-f017:**
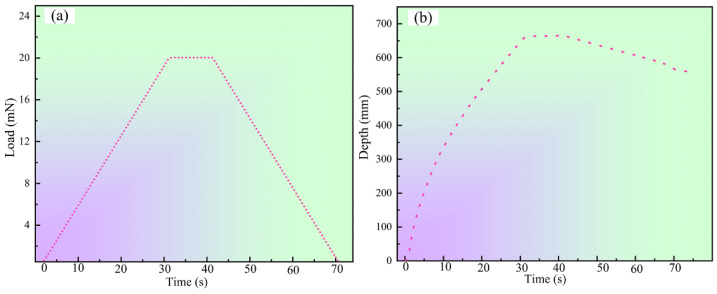
(**a**) Force change curve. (**b**) Variation in indentation depth.

**Figure 18 micromachines-14-02106-f018:**
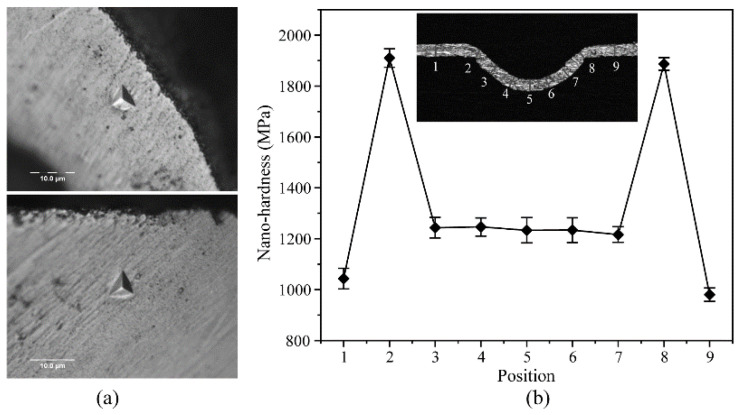
The change in hardness curve. (**a**) Microgroove cross section nanohardness indentation morphology; (**b**) The nano-hardness values measured at the formed microgroove cross-section.

**Table 1 micromachines-14-02106-t001:** Factors and levels of the response surface methodology.

		Level		
Factor	Mark	−1	0	1
*H* (mm)	X_1_	2	3	4
Energy (mJ)	X_2_	23	27	31
Number of impacts	X_3_	1	3	5

**Table 2 micromachines-14-02106-t002:** Experiment design and results of the response surface methodology.

	Position of Bubble (mm)	Energy (mJ)	Number of Impacts	Depth (μm)
1	2	27	1	100
2	4	27	1	60
3	3	27	3	102
4	4	31	3	103
5	3	27	3	102
6	4	23	3	95
7	3	27	3	102
8	3	31	5	105
9	2	23	3	104
10	3	27	3	102
11	2	27	5	105
12	3	23	1	72
13	4	27	5	104
14	3	31	1	103.5
15	2	31	3	105
16	3	27	3	102
17	3	23	5	100

**Table 3 micromachines-14-02106-t003:** Model variance analysis.

Source	Sum of Squares	df	Mean Square	F Value	*p*-Value	
Model	2236.67	9	248.52	6.83	0.0095	significant
*X* _1_	338.00	1	338.00	9.29	0.0186	
*X* _2_	258.78	1	258.78	7.12	0.0321	
*X* _3_	770.28	1	770.28	21.18	0.0025	
*X* _1_ *X* _2_	12.25	1	12.25	0.3369	0.57	
*X* _1_ *X* _3_	380.25	1	380.2	10.46	0.0144	
*X* _2_ *X* _3_	175.56	1	175.5	4.83	0.0640	
*X* _1_ ^2^	10.28	1	10.28	0.2827	0.6114	
*X* _2_ ^2^	7.25	1	7.25	0.1995	0.6687	
*X* _3_ ^2^	282.25	1	282.25	7.76	0.0271	
Residual	254.56	7	36.37			
Lack of Fit	254.56	3	84.85			
Cor Total	2491.24	16				

## Data Availability

Data available in a publicly accessible repository that does not issue DOIs.

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
