# Peer review of "Experiment and Simulation Study of the Laser-Induced Cavitation Bubble Technique for Forming a Microgroove in Aluminum Foil"

_micromachines, 2023, doi:10.3390/mi14112106_

Round 1
Reviewer 1 Report
Comments and Suggestions for Authors
In this work laser-induced cavitation bubble stamping process was studied using various methods. In my point of view, the novelty of the paper was not so obvious, or the difference between the previous studies should be stressed. The other comments are as follows:
(1) How was the position of the cavitation bubbles determined?
(2) How was the microgroove produced by the proposed method measured? The meaning of the width should be detailed.
(3) The relation between the surface quality with impact numbers may be discussed.
(4) In which conditions the forming shape could be the same with the prefabricated mold?
(5) I am wondering if the hardness could be controlled by adjusting the shape of the micro cavity in the mold.
(6) The conclusion section may be improved.
Comments on the Quality of English LanguageIn this work laser-induced cavitation bubble stamping process was studied using various methods. In my point of view, the novelty of the paper was not so obvious, or the difference between the previous studies should be stressed. The other comments are as follows:
(1) How was the position of the cavitation bubbles determined?
(2) How was the microgroove produced by the proposed method measured? The meaning of the width should be detailed.
(3) The relation between the surface quality with impact numbers may be discussed.
(4) In which conditions the forming shape could be the same with the prefabricated mold?
(5) I am wondering if the hardness could be controlled by adjusting the shape of the micro cavity in the mold.
(6) The conclusion section may be improved.
Author Response
For review article
Response to Reviewer 1 Comments
Thank you very much for taking the time to review this manuscript. Below is our response to the comments:
Comments 1: In this work laser-induced cavitation bubble stamping process was studied using various methods. In my point of view, the novelty of the paper was not so obvious, or the difference between the previous studies should be stressed. The other comments are as follows: (1) How was the position of the cavitation bubbles determined?
Response 1: Thank you for your comments, the position of the bubble is determined by the laser focus.
Comments 2: How was the microgroove produced by the proposed method measured? The meaning of the width should be detailed.
Response 2: Thank you for your comments. A laser confocal microscope (OLS4100, Olympus, Germany) was employed to measure the width at the entrance of the microgroove created by laser-induced cavitation bubble impact.
Comments 3: The relation between the surface quality with impact numbers may be discussed.
Response 3: Thank you for your professional comments. The impact numbers have a significant impact on surface quality, and the relationship between the two has been studied in previous papers (https://doi.org/10.1016/j.jmapro.2022.04.019).
Comments 4: In which conditions the forming shape could be the same with the prefabricated mold?
Response 4: Thank you for your constructive comments. It is difficult to replicate the shape of a mold, and many scholars have attempted to process micro holes, but have not fully replicated the shape of the mold. (https://doi.org/10.1016/j.optlastec.2019.02.048; https://doi.org/10.1016/j.jmapro.2021.05.010). In generally, the possible reason is that the impact force generated by cavitation is uneven and the interaction time with the workpiece is relatively short. When a material undergoes plastic deformation at the micro scale, size effects occurring and affecting the deformation of the workpiece. It has a great scientific significance and research value and we will be studied in the near future.
Comments 5: I am wondering if the hardness could be controlled by adjusting the shape of the micro cavity in the mold
Response 5: Thank you for your comment. In general, the change in hardness is mostly attributed to the microscopic deformation of the material at different locations after an impact force. Accordingly, by adjusting the shape of the mold, it is possible to change the local microscopic deformation of the material in theory, and so may change the hardness. As a consequence, we will deeply investigate the influence of the shape of the mold on the hardness characteristics.
Comments 6: The conclusion section may be improved
Response 6: Thank you for your comment, revised as suggested (see line 382 to 385, and line 389 to 390).
I trust the revised paper would satisfy your requirements.
Look forward to your reply in due course.
Yours sincerely,
Zhixiang Zou
Reviewer 2 Report
Comments and Suggestions for Authors
Authors both numerically and experimentally investigate the evolution of a bubble oscillating near the guide hole. They discussed the suitable laser and focusing parameters for the optimization of laser-induced cavitation bubble stamping microgrooves. The experimental studies and results obtained (both in the experiment and numerically) are described in detail and flawlessly.
However, in my opinion, section 3.2 (Governing equations) lacks clarity. A few things should be specified and clarified here: 1) what equations were addressed for the numerical solution; 2) why here is limited only to the equations of volume and density; 3) the text does not indicate the meanings of all the notations used in the equations (1-3); 4)What number of dimensions were used to solve equations?
Author Response
For review article
Response to Reviewer 2 Comments
Thank you very much for taking the time to review this manuscript. Below is our response to the comments:
Comments 1: Authors both numerically and experimentally investigate the evolution of a bubble oscillating near the guide hole. They discussed the suitable laser and focusing parameters for the optimization of laser-induced cavitation bubble stamping microgrooves. The experimental studies and results obtained (both in the experiment and numerically) are described in detail and flawlessly. However, in my opinion, section 3.2 (Governing equations) lacks clarity. A few things should be specified and clarified here:
What equations were addressed for the numerical solution.
Response 1: Thank you for your comments, The mass continuum equation, the Navier-Stokes equation, and the interface equation were used to solve the numerical solution.
Comments 2: Why here is limited only to the equations of volume and density.
Response 2: Thank you for your comments, these equations are also related to mass, viscosity, etc.
Comments 3: The text does not indicate the meanings of all the notations used in the equations (1-3).
Response 3: Thank you for your constructive comments, all the notations has been added in the revised manuscript (see line 143, and line 145).
Comments 4: What number of dimensions were used to solve equations?
Response 4: Thank you for your comments, the 3 number of dimensions were used to solve equations.
I trust the revised paper would satisfy your requirements.
Look forward to your reply in due course.
Yours sincerely,
Zhixiang Zou